# Using a Bodily Weight-Fat Scale for Cuffless Blood Pressure Measurement Based on the Edge Computing System

**DOI:** 10.3390/s24237830

**Published:** 2024-12-07

**Authors:** Shing-Hong Liu, Bo-Yan Wu, Xin Zhu, Chiun-Li Chin

**Affiliations:** 1Department of Computer Science and Information Engineering, Chaoyang University of Technology, Taichung 41349, Taiwan; shliu@cyut.edu.tw (S.-H.L.); s11327606@gm.cyut.edu.tw (B.-Y.W.); 2Department of AI Technology Development, M&D Data Science Center, Institute of Integrated Research, Institute of Science Tokyo, Tokyo 101-0062, Japan; zhu.xin@tmd.ac.jp; 3Department of Automatic Control Engineering, Feng Chia University, Taichung 40724, Taiwan

**Keywords:** blood pressure measurement, ballistocardiogram, impedance plethysmogram, bodily weight-fat scale, edge computing

## Abstract

Blood pressure (BP) measurement is a major physiological information for people with cardiovascular diseases, such as hypertension, heart failure, and atherosclerosis. Moreover, elders and patients with kidney disease and diabetes mellitus also are suggested to measure their BP every day. The cuffless BP measurement has been developed in the past 10 years, which is comfortable to users. Now, ballistocardiogram (BCG) and impedance plethysmogram (IPG) could be used to perform the cuffless BP measurement. Thus, the aim of this study is to realize edge computing for the BP measurement in real time, which includes measurements of BCG and IPG signals, digital signal process, feature extraction, and BP estimation by machine learning algorithm. This system measured BCG and IPG signals from a bodily weight-fat scale with the self-made circuits. The signals were filtered to reduce the noise and segmented by 2 s. Then, we proposed a flowchart to extract the parameter, pulse transit time (PTT), within each segment. The feature included two calibration-based parameters and one calibration-free parameter was used to estimate BP with XGBoost. In order to realize the system in STM32F756ZG NUCLEO development board, we limited the hyperparameters of XGBoost model, including maximum depth (max_depth) and tree number (n_estimators). Results show that the error of systolic blood pressure (SBP) and diastolic blood pressure (DBP) in server-based computing are 2.64 ± 9.71 mmHg and 1.52 ± 6.32 mmHg, and in edge computing are 2.2 ± 10.9 mmHg and 1.87 ± 6.79 mmHg. This proposed method significantly enhances the feasibility of bodily weight-fat scale in the BP measurement for effective utilization in mobile health applications.

## 1. Introduction

Recently, the lifespans of the world’s population are increasing, and society is gradually aging. According to the report of the United Nations [1], the number of elderly people (over 65) in the world in 2019 was 703 million, and this is estimated to double to 1.5 billion by 2050. From 1990 to 2019, the proportion of the global population over 65 years old increased from 6% to 9%, and the proportion of the elderly population is expected to further increase to 16% by 2050. Thus, the cost of home care for elders will significantly increase. In home care, the monitoring of elderly people living alone is a major issue. Some studies have proposed that engaged elders could use sophisticated digital technologies for self-monitoring and self-care [2]. Large amounts of data and information are measured from such digital apparatuses, relating to the past, present, and future physical and mental health or condition of an individual, which can be analyzed to extract knowledge for improving public health and providing basic services [3].

The bodily weight-fat scale and blood pressure (BP) monitor are the most popular apparatuses used in the home environment [4]. The study of Lacey et al. showed that the usual systolic blood pressure (SBP) of people ages 40–79 years in China was continuously and positively associated with vascular diseases throughout the range of 120–180 mmHg, with each 10 mmHg higher than usual; SBP was associated with an approximately 30% higher risk of ischemic heart disease [5]. Neter et al. performed a meta-analysis of hypertension and bodily weight and found that the SBP and diastolic blood pressure (DBP) would reduce by 1.05 mmHg and 0.92 mmHg when expressed per kilogram of weight loss [6]. Hypertension during pregnancy was associated with an increased risk of subsequent cardiovascular disease and arterial hypertension, such as ischemic heart disease, myocardial infarcts, kidney disease, and diabetes mellitus [7]. The BP control could be achieved to slow the progression of renal damage [8]. Thus, BP monitoring, risk factor evaluation, and early intervention could benefit elders for self-management interventions of chronic disease. Moreover, a diet rich in fruits, vegetables, low-fat dairy products, fiber, and minerals, would produce a potent anti-hypertensive effect [9]. Many studies show that there is a positive relationship between being overweight or obese and BP with the risk of hypertension [10]. Furthermore, weight management is extremely important for alerting the risks related to abdominal adiposity and its complications [11]. However, sudden weight loss may also indicate the dangers of sarcopenia, the loss of skeletal muscle mass combined with low muscle function, and the increased risk of hip fracture [12]. Considering the association between reduced mortality and increased body mass index in older adults, daily weight measurement may enable successful weight loss interventions including physical activity and nutritional interventions for older adults. Thus, if the bodily weight-fat scale has the function of BP measurement, people will easily monitor their weight and BP every day.

The cuffless BP measurement has been studied for the past 10 years since Sharwood-Smith et al. in 2006 found a significant relation between the pulse transit time (PTT) and the change in BP under anesthesia [13]. PTT generally is defined as the transmitting time of a pulse wave from the aortic arch to the peripheral vessel, which can be measured by electrocardiogram (ECG) and photoplethysmogram (PPG) [13]. The fundamental of this measurement is the Moens-Korteweg equation [14]. Liu et al. proposed a novel method of BP measurement which detected PTT with the ballistocardiogram (BCG) measured by a weight scale and PPG measured at the toe [15]. Next year, they used the bodily weight-fat scale to perform the cuffless BP measurement [16]. However, the parameters of PTT were extracted by the manual process, and the linear regression algorithm was used to estimate the BP. Thus, the Pearson correlation coefficient (PCC) and root mean square error (RMSE) for SBP and DBP were only 0.754 vs. 0.533, and 7.3 ± 2.1 mmHg vs. 4.5 ± 1.8 mmHg.

An edge computing system combines 5G communication and modern computing techniques, which can perform real-time monitoring or measurement in a mobile device. Zheng et al. studied the wireless-powered multi-access edge computing network, where wireless devices conducted either local computing or task offloading for their undividable computation tasks [17]. Now, some studies reviewed its approaches, opportunities, and challenges in smart health [18,19]. The modern computing techniques focus on deep learning and ensemble machine learning algorithms, which can be executed in a microcontroller. Chin et al. used an Arduino Nano 33 BLE Sense development board to classify the emergency vehicle sirens with an EfficientNet-based ensemble model [20]. Rahman et al. developed a deep learning model to detect COVID-19 symptoms based on a smartphone [21]. Goossens et al. proposed state-of-the-art algorithms of edge computing for real-time BP estimation and ECG compression [22]. However, this study focused on power consumption and execution time. Many studies proposed the methods of cuffless BP measurement but few researchers studied the cuffless BP measurement in an edge computing environment. Bernard et al. used a cloud server to collect the PPG signal measured by different bedside monitors and built the deep learning and machine learning models to estimate BP [23]. Sun et al. also used the PPG signal from the MIMIC-IV database to estimate BP with five convolutional neural networks in the Arduino Nano 33 BLE development board. Only AlexNet had the least loss in performance at about 8%. [24]. Ahmed et al. also used the MIMIC-IV database to build an edge computing system with six machine learning models in the ESP32 Wrover Board [25]. The mean absolute error (MAE) of SBP and DBP were 14.08 ± 17.82 mmHg and 6.85 ± 9.16 mmHg, respectively. However, the edge computing technique should include the sensors, signal measurement, signal process, feature extraction, building model, adjustment of hyperparameters, imbedding in a microcontroller, and evaluation of performance. The previous studies did not fully fit the requirements of the edge computing system.

The aim of this study is to show how a cuffless BP measurement system is realized in an edge computing environment. The BCG and impedance plethysmogram (IPG) were measured from a bodily weight-fat scale when a user was standing on it. Signals were filtered and segmented to extract the calibration-based and calibration-free PTTs according to a previous study [16]. Then, we explored the XGBoost to estimate BP in server-based computing. The hyperparameters of the model would be limited to fit the resources of the STM32F756ZG NUCLEO (STMicroelectronics NV, Plan-les-Ouates, Geneva, Switzerland) development board, and also keep its performance. Finally, we compared the performance of server-based computing with edge computing.

The main contributions of this study are summarized as follows:This study uses the self-made circuits to measure BCG and IPG signals from the bodily weight-fat scale. These signals were filtered and segmented to extract the calibration-free and calibration-based PTT parameters.This study proposes an operation procedure to extract these PTT parameters in the edge computing system.Our proposed models for SBP and DBP estimations in this study are tested on the STM32F756ZG NUCLEO development board. This system could perform the cuffless BP measurement in real time.This study verifies the performances of server-based computing and edge computing. Edge computing has a loss in performance at about 8%.

## 2. Materials and Methods

Figure 1 shows the flowchart of this study. BCG and IPG signals were measured from a bodily weight-fat scale, which were filtered by the infinite impulse response filters to remove the baseline and noise. The signals were segmented for the feature extraction. In the training phase, all samples were used to build a model in server-based computing to estimate BP. Then, the hyperparameters of the model would be limited to let the memory size fit the resources of the STM32F756ZG NUCLEO development board. In the testing phase, the signals were fed into this edging computing system to estimate BP and show them in a graphic liquid crystal display (LCD) in real time. The sampling rate is 500 Hz.

### 2.1. BCG and IPG Measurement

We designed the circuits to measure BCG and IPG signals from the strain gauges and electrode pads of the bodily weight-fat scale. Figure 2 shows the BCG schematic which used two strain gauges to detect the BCG signal. U1 (AD620, Analog Device, Norwood City, OH, USA) is an instrument amplifier, U2A (TL082, Texas Instruments, Dallas City, TX, USA) and U4A (TL082) are two second-order Butterworth high pass filters, where the cutoff frequency is 0.5 Hz, and U3 (TL082) and U4B are a fourth-order Butterworth low pass filter, where the cutoff frequency is 20 Hz. U2B and U5A (TL082) are two noninverted amplifiers, which all gain 500. U5B is a baseline shift, whose output is the BCG signal. Figure 3 shows the IPG schematic which used four electrode pads to feed the fixed alternated current into the body and extract the potential voltage of body. An oscillation frequency (32 kHz) is generated by the crystal of the bodily weight-fat scale, which is inputted into the circuit (U1A, TL082) of the fixed current source and demodulator (U3, AD835). U9A (TL082) is a voltage follower. The fixed current is inputted into the first pad. The fourth pad is the ground. The potential voltage is extracted from the second and third pads. U2 (AD620) is an instrument amplifier (AD620), and U1B is a second-order Butterworth high pass filter (0.3 Hz of cutoff frequency) to remove the baseline voltage, which is demodulated by U3. Then, U4A (TL082), U5A (TL082), and U6A (TL082) are three second-order Butterworth low pass filters, where the cutoff frequency is 10 Hz. U4B and U5B are the noninverted amplifiers, which all gain 500. U6B is also a second-order Butterworth high pass filter. U7A is a two-pole notch filter, where the stop frequency is 60 Hz. U7B is a baseline shift, where output is the IPG signal.

### 2.2. Signal Processing and Parameter Extraction

The infinite impulse response filters were used to remove the noises of BCG and IPG. We used the recursive approach to filter signals in real time. For the BCG signal, Equation (1) represents the Z transfer function of the fourth-order Butterworth high pass filter (0.5 Hz of cutoff frequency), and Equation (2) represents the Z transfer function of the fourth-order Butterworth low pass filter (20 Hz of cutoff frequency). For IPG signal, Equation (3) represents the Z transfer function of the second-order Chebyshev high pass filter (0.3 Hz of cutoff frequency), and Equation (4) represents the Z transfer function of the fourth-order Chebyshev low pass filter (10 Hz of cutoff frequency).
(1)$$H\left(z\right)=\frac{0.98+\left(-1.97\right)z^{-1}+0.98z^{-2}}{1+\left(-1.99\right)z^{-1}+0.99z^{-2}},$$
(2)$$H\left(z\right)=\frac{1.14e^{-5}+4.56e^{-5}z^{-1}+6.84e^{-5}z^{-2}+4.56e^{-5}z^{-3}+1.14e^{-5}z^{-4}}{1+\left(-3.75\right)z^{-1}+5.31z^{-2}+\left(-3.35\right)z^{-3}+0.79z^{-4}}$$
(3)$$H\left(z\right)=\frac{0.99+\left(-3.96\right)z^{-1}+5.95z^{-2}+\left(-3.96\right)z^{-3}+0.99z^{-4}}{1+\left(-3.98\right)z^{-1}+5.95z^{-2}+\left(-3.95\right)z^{-3}+0.98z^{-4}}$$
(4)$$H\left(z\right)=\frac{1.83e^{-4}+7.32e^{-4}z^{-1}+1.1e^{-3}z^{-2}+7.32e^{-4}z^{-3}+1.83e^{-4}z^{-4}}{1+\left(-3.34\right)z^{-1}+4.23z^{-2}+\left(-2.4\right)z^{-3}+0.51z^{-4}}.$$

The filtered signals were segmented with 1024 points of window and 512 points of overlap. Thus, there were 22,620 segments. We chose 2262 segments as samples as the signals were of good quality with the manual method. According to ref. [16], PTT was extracted from BCG and differential IPG (dIPG) signals, which was defined as the time between the J wave of BCG and main peak of dIPG. We assumed the range of heat rate between 60 beats/minute (BPM) to 200 BPM. Because the window is 2 s and sampling rate is 500 Hz, there are at least one to six PTT values in a segment. Figure 4 shows the flowchart of PTT measurement. dIPG is the differential IPG, and ddIPG is the second differential IPG. There are three parts, first finding the peak of IPG (IPG_P), the second finding the peak of dIPG, and the third finding the J wave of BCG (BCG_Peak). In the first part, all IPG_P of IPG are detected from the zero-crossing points of dIPG, then the first three largest peaks are averaged as the Threshold. If peaks are larger than the Threshold, they will be marked as the T_IPG_Peak. T_IPG_Peak would be verified as the peak (V_IPG_Peak) within 25 points before and after. The interval between two V_IPG_Peaks is captured. Cycle_Average is the mean of all intervals, which is used to determine the truth IPG_P. In the second part, all foots of dIPG (dIPG_F) are detected from the zero-crossing points of ddIPG, which also are verified as the foot (V_dIPG_Foot) within 20 points before and after. These verified foots are closed as the truth foots of dIPG by IPG_P and Cycle_Average. Truth IPG_P searches forward to find the truth foot of dIPG (Truth dIPG_F) within the range of Cycle_Average multiplied by 0.7. The peak of dIPG (dIPG_P) is found from truth dIPG_F searching backward. In the third part, the differential BCG (dBCG) is obtained from BCG. All peaks (BCG_P) of BCG are detected from the zero-crossing points of dBCG. Truth BCG_P is the largest value between two dIPG_P. Because BCG_P is a phase lead of dIPG_P, PTT(i) is defined below:(5)PTT(i)=dIPG_P(i)−BCG_P(i) ,
where i is the number of beat-to-beat in a segment. Thus, the calibration-free PTT is defined as the average of all PTT(i). Calibration-based PTT parameters, PTT_SYS_ and PTT_DIA_, are the calibration-free PTT normalized by the SBP and DBP in the resting status.

### 2.3. Models of Blood Pressure Estimation

This study employed the prevalent nonlinear regression model, XGBoost, developed by Python 3.7. The server-based computing environment consists of an Intel Core i7-8700 CPU and an GeForce GTX3070 GPU (Nvidia Corporation, Santa Clara, CA, USA). The edge computing system is the STM32F756ZG-NUCLEO development board designed by Advanced RISC Machines Ltd., and produced by STMicroelectronics, in which the development environment is STM32CubeIDE 1.7.0. where the microcontroller of the board uses ARM^®^ Cortex^®^-M7 core with a high-performance 32-bit, which has 1 MB Flash and 340 KB RAM.

#### 2.3.1. XGBoost

XGBoost (eXtreme Gradient Boosting) represents an enhanced version of the Gradient Boosting technique, combining numerous weak decision trees to construct a powerful predictive model. It conducts feature splitting to grow each tree, with each new tree representing a new function aimed at fitting the residual of the previous prediction. Once N trees are generated, the model predicts the score of a sample based on its characteristics, with each tree directing the sample to a corresponding leaf node, each of which holds a score. The total scores from all trees determine the predicted value of the samples. Compared to conventional classification and regression techniques, XGBoost typically demonstrates superior accuracy due to its robustness and adaptive learning capability [26]. For this study, the learning rate was set to 0.0497, the maximum depth to 12, and the number of trees to 100.

#### 2.3.2. Python to C Codes

In a server-based computing system, the format of the trained model could be .pb, .onnx, kept, tflite, or .h5. However, these formats cannot be directly imported into the microcontroller, so a special conversion of the format is required. When training the model, we use the m2cgen (Model 2 Code Generator) function library to convert the XGBoost model as C code, as shown in Figure 5. The last three comments convert the estimation models of SBP and DBP as C-code models. First, we import the m2cgen tool, “import m2cgen as m2c”. Then, we export C-code models of SBP and DBP estimation as SBP_code and DBP_code, “SBP_code = m2c.export_to_c(SBP_model)” and “DBP_code = m2.export_to_c(DBP_model)”.

### 2.4. Data Collection

There were 17 subjects (11 males and 6 females) participating in this study, whose ages were 20.2 ± 1.1 years (from 22 to 19 years of age), weights were 62.8 ± 16.1 kg (from 115 to 43 kg), and heights were 166.1 ± 8.0 cm (from 186 to 152 cm). This experiment was approved by the Research Ethics Committee of Chung Shan University Hospital (No. CS2-21194), in Taichung City, Taiwan.

When subjects were resting for about five minutes, they filled out an informed consent form to confirm their participation in this experiment. All subjects did not have any cardiac disease. Their BPs were measured by a digital sphygmomanometer (HM-7320, Omron, Osaka, Japan), which served as the reference BP. The cuff was wrapped on the left upper arm. A commercial bodily weight-fat scale (HBF-371, Omron, Osaka, Japan) which was modified by adding self-made circuits was used to measure the BCG and IPG signals. The experiment procedure is mentioned below.

Subjects stood on the bodily weight-fat scale for five minutes to measure IPG and BCG signals, with blood pressure measured once. This BP was the baseline, which was used to normalize PTT.Subjects ran on a treadmill at a fixed speed for a minimum of six minutes to boost SBP above 20 mmHg from the resting SBP. If the SBP is less than 20 mmHg, subjects continually ran about 1 min.After treadmill exercise, subjects stood on the bodily weight-fat scale for six minutes to measure IPG and BCG signals. Blood pressure was concurrently measured every minute. Thus, there were six times for BP measurements.Subjects underwent the procedure four times and rested at least one week between two experiments.

We hypothesized that BP would drop after the exercise while standing on the bodily weight-fat scale. Thus, we utilized the linear interpolation method to determine the reference BP for each pulse between two BP measurements. The reference BP of the segment was the average of all pulses. According to the IEEE Standard for Wearable, Cuffless Blood Pressure Measuring Devices [27], BPs were defined as four categories, normal BP (SBP < 120 mmHg and DBP < 80 mmHg), prehypertension (SBP between 120 mmHg and 129 mmHg and DBP < 80 mmHg), stage_1 hypertension (SBP between 130 mmHg and 139 mmHg or DBP between (80 mmHg and 89 mmHg), and stage_2 hypertension (SBP > 140 mmHg or DBP > 90 mmHg). The numbers of samples for the normal BP, prehypertension, stage_1 hypertension, and stage_2 hypertension were 597, 182, 590, and 893. In each category, the ratio between training and testing samples was 4:1. 

### 2.5. Statistical Analysis

The PCC, *ρ*, is used to establish the relationship between the target and estimated BPs. Equation (6) shows the correlation coefficient,
(6)$$\rho \left(X,Y\right)=\frac{\sum_{i=1}^{n}\left(x_{i}-\bar{x}\right)\left(y_{i}-\bar{y}\right)}{\sqrt{\sum_{i=1}^{n}{\left(x_{i}-\bar{x}\right)}^{2}}\sqrt{\sum_{i=1}^{n}{\left(y_{i}-\bar{y}\right)}^{2}}},$$
where *x_i_* and *y_i_* are the target and estimated BPs, $\bar{x}\;\mathrm{and}\;\bar{y}$ are the mean of target and estimated blood pressures, and n is the number of testing samples. The quantitative analysis is expressed as the mean ± standard deviation. Computations of mean error (ME) and mean absolute error (MAE) were performed to evaluate discrepancies between the target and predicted values when using the test data. The ME and MAE are delineated in Equations (7) and (8).
(7)$$ME=\frac{1}{n}\sum_{i=1}^{n}(y_{i}-x_{i}),$$
(8)$$MAE=\frac{1}{n}\overset{n}{\underset{i=1}{\sum }}\left|y_{i}-x_{i}\right|.$$

## 3. Results

We utilized server-based computing to evaluate the performance of XGBoost with the different number of trees according to PCC, ME, and MAE. Then, the optimal models for SBP and DBP estimations were selected and converted to C-code models. The models were imbedded in the STM32F756ZG-NUCLEO development board. Finally, we showed the real performances in the edge computing system.

### 3.1. Analysis of Server-Based Computing

We used 5-fold cross-validation to evaluate the PCCs of XGBoost with the different numbers of trees for SBP and DBP estimations, as shown in Table 1. Then, the *t*-test was used to analyze the differences in models between 100 trees and the other trees. We did not find any differences (all *p*-values > 0.05). Because the memory size of XGBoost with 100 trees was 1.6 Mbytes, this model could not be imbedded in STM32F756ZG-NUCLEO development board. For SBP and DBP estimation, XGBoost with 60 trees had the best PCCs, 0.79307 ± 0.015 and 0.78251 ± 0.015. Thus, we analyzed the performances of XGBoost with the smaller sizes. Table 2 shows the MAEs of SBP and DBP estimations with the different numbers of trees, excluding 100 trees. We found that the model’s larger size (more trees), estimated BP less than MAE. The smallest MAEs of SBP and DBP estimations are 7.52 ± 0.18 mmHg and 4.88 ± 0.11 mmHg at XGBoost using 90 trees.

### 3.2. Analysis of Edge Computing

The STM32F756ZG microcontroller only has 1 Mbytes of flash memory. We chose models with 80 and 70 trees to convert as C-code models and imbedded them in this microcontroller. Figure 6 shows the sizes of models in the flash memory. When models are 80 trees, the size of models for estimating SBP and DBP are 339.77 Kbytes and 350.3 Kbytes, respectively, as shown in Figure 6a. When models are 70 trees, the size of models for estimating SBP and DBP are 240.36 Kbytes and 276.68 Kbytes, respectively, as shown in Figure 6b. The testing signals, BCG and IPG, were inputted to the STM32F756ZG NUCLEO development board by UART (Universal Asynchronous Receiver/Transmitter) of a personal computer (PC). The resolution of analog-to-digital conversion (ADC) was 12 bits. Thus, the transmission rate of data was 2k bytes/second to simulate the 500 Hz sampling rate. Table 3 shows the metrics of server-based computing and edge computing when the trees of models were 80. In edge computing, PCCs of SBP and DBP estimations are only 0.73 and 0.78, which are lower than PCCs in server-based computing, 0.80 and 0.81. ME and MAE for SBP estimation in edge computing increase to 2.2 ± 10.9 mmHg and 8.58 ± 7.2 mmHg, compared to server-based computing, 2.64 ± 9.71 mmHg and 7.63 ± 0.20 mmHg. ME and MAE for DBP estimation in edge computing increase to 1.87 ± 6.79 mmHg and 5.27 ± 4.66 mmHg, compared to server-based computing, 1.52 ± 6.32 mmHg and 5.01 ± 0.11 mmHg.

## 4. Discussions

Because the amount of data measured from wearable, portable, or mobile devices requires a fast, real-time, and secure process, edge computing emerged at this historic moment [28]. Zha et al. [29] described the concept of edge computing, “Edge computing is a new computing model that unifies resources that are close to the user in geographical distance or network distance to provide computing, storage, and network for applications service”. Cao et al. [30] proposed the shortcomings of cloud-based big data processing, including real-time requirements, data privacy and security, and energy consumption. Thus, they thought that edge computing technology provides artificial intelligence services for rapidly growing terminal devices and data and makes services more stable. Therefore, edge computing has proximity and location awareness, and provides users with near-end services. In this study, we utilized an STM32F756ZG NUCLEO development board for BP measurement based on the bodily weight-fat scale. When users are standing on a weight-fat scale, they not only measure bodily weight and fat percentage, but also measure the BP. The advantage of this study is using an STM32F756ZG microcontroller to realize the cuffless BP measurement, which does not need the operating system and has a low power model. Thus, this system can work with a carbon zinc battery. However, this microcontroller has little memory, only 1Mbytes. Thus, the more complex model cannot be executed in it. 

Now, many traditional medical apparatuses have been developed as mobile or wearable devices, such as ECG patches [31], ECG and SpO2 watches [32,33], noninvasive blood-glucose measurement [34], and electromyogram (EMG) patches [35]. In these devices, a few devices have received approval for the product. Many devices are only a prototype because their accuracy and precision do not arrive at the level of standard. How to utilize machine learning and deep learning algorithms to improve the performance of wearable devices has been a popular search project now. The cuffless BP measurement has been studied for decades, and which major challenge also is its accuracy. Pandit et al. reviewed the promises and challenges of cuffless BP monitors [36]. In this literature, there were 31 papers using machine learning and deep learning algorithms for BP measurement. However, these studies all were on the server-based computing environment. In this study, we proposed edge computing technology to realize the cuffless BP measurement, and also compared the performance between server-based computing and edge computing as shown in Table 3. The metrics of edge computing all are lower than server-based computing. The major reason is the resolution. STM32F756ZG is a 32-bit microcontroller. The digital filters were designed by MATLAB^®^. The coefficients of filters in edge computing were not the same values designed by MATLAB^®^. Thus, the determined PTT also has some errors. Liu et al. also discussed this problem of the digital signal process in EMG patches [35]. Moreover, XGBoost is an ensemble machine learning algorithm, which has the accumulated error in the last leaves.

Tan et al. evaluated the ability of a commercially available cuffless wearable device, Aktiia, to track 24 h blood pressure with a conventional ambulatory blood pressure monitor [37]. The mean bias and limits of agreement for 24 h SBP and DBP were 5.3 [−14.4; 25.0] mmHg and 4.7 [−7.2; 16.6] mmHg. Liu et al. used the smartwatch to perform a large-scale validation study in cuffless BP measurement [38]. They measured ECG and PPG signals and used a machine learning algorithm. The best-performing calibration-based model yielded estimation errors of 2.31 ± 9.57 mmHg for SBP and 1.33 ± 6.43 mmHg for DBP. In this study, although MEs of estimated BP in edge computing (SBP: 2.2 ± 10.9 mmHg, DBP: 1.87 ± 6.79 mmHg) were lower than the results of Liu et al., the errors in server-based computing (SBP: 2.64 ± 9.71 mmHg, DBP: 1.52 ± 6.32 mmHg) were very close to their results. According to the British Hypertension Society (BHS) standard, SBP is C grade and DBP is A grade [27]. Moreover, previous studies used the statistic BP measurement and used the cuff BP monitor to measure the reference BP at the same time. In our study, we measured the dynamic BP, and only measured reference BP every minute.

Table 4 presents a comparative analysis of our method with other studies that utilized either ECG or BCG as the proximal reference, and IPG or PPG as the distal reference for cuffless blood pressure measurement. RMSE, ME, and PCC are the metrics. Notably, our proposed method exclusively relied on BCG and IPG data obtained from the weight-fat scale and demonstrated superior performance compared to previous studies. Our proposed method has the best PCCs of 0.80 and 0.81, and better MEs of 2.64 ± 9.71 mmHg and 1.52 ± 6.32 mmHg of ME for SBP and DBP estimations.

## 5. Conclusions

In this study, the innovation of the proposed approach is to use the bodily weight-fat scale for the real-time cuffless BP measurement based on the STM32F756ZG NUCLEO development board. The edge computing has only an 8% loss in performances of server-based computing. In order to address the real-time measurement, the self-made circuits were used to measure BCG and IPG signals from the bodily weight-fat scale. These signals were filtered and segmented. We proposed a method to extract PTT from BCG and IPG signals in real time. Then, we limited the hyperparameters of XGBoost to reduce the memory size of models for SBP and DBP estimations which could be executed on an STM32F756ZG NUCLEO development board. In the future, if the calibration-based or calibration-free parameters of PTT can be improved, the accuracy of real-time cuffless BP measurement will arrive at the A grade for the IEEE Standard for Wearable, Cuffless Blood Pressure Measuring Devices. This advancement would support convenient BP monitoring in daily life and facilitate the progress of mobile health.

## Figures and Tables

**Figure 1 sensors-24-07830-f001:**
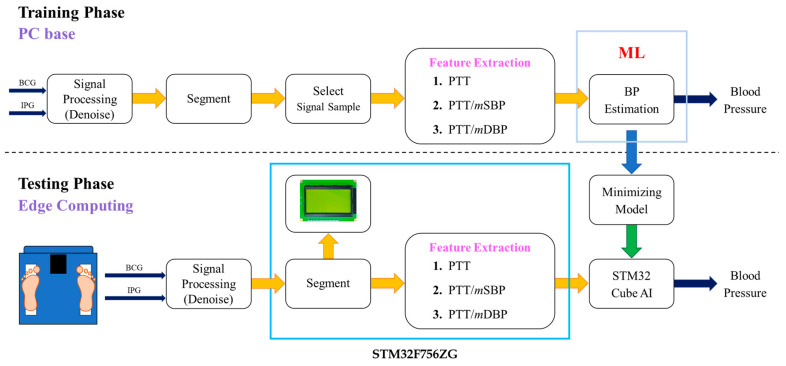
The flowchart of this study includes the training phase in server-based computing and testing phase in edge computing.

**Figure 2 sensors-24-07830-f002:**
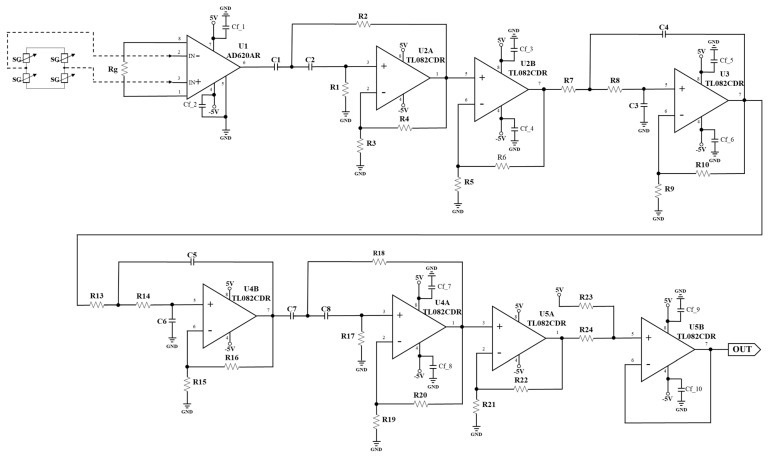
The self-made BCG schematic.

**Figure 3 sensors-24-07830-f003:**
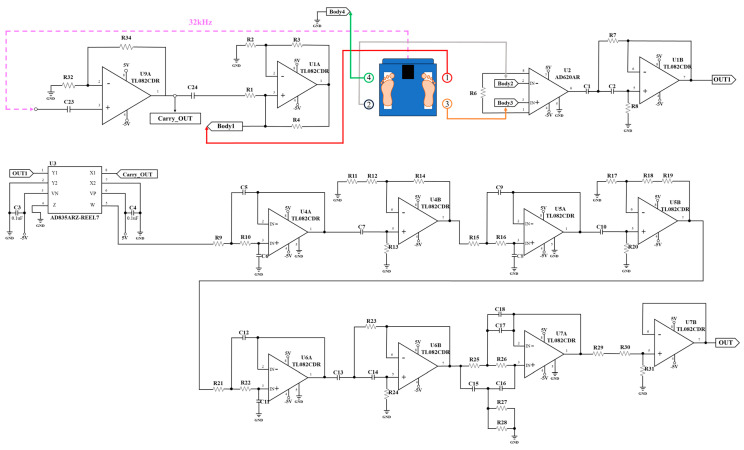
The self-made IPG schematic.

**Figure 4 sensors-24-07830-f004:**
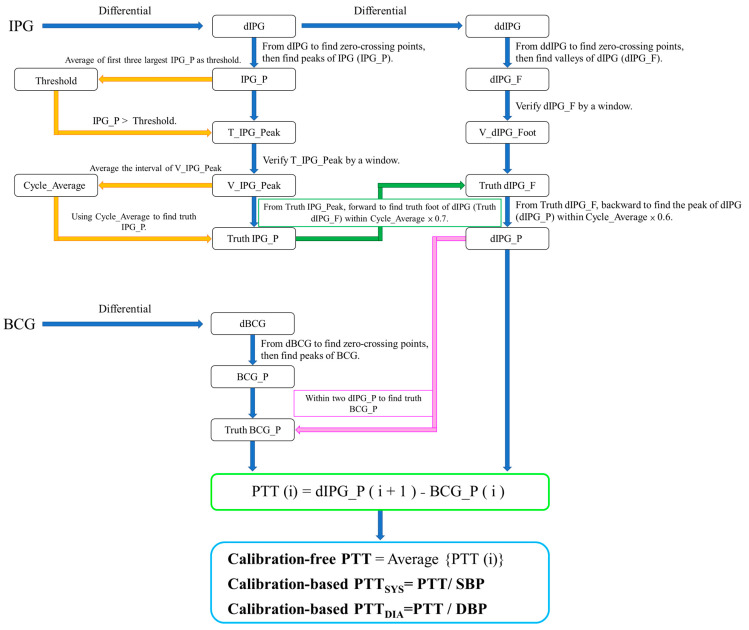
The flowchart of calibration-free PTT and calibration-based PTT measurement from BCG and IPG.

**Figure 5 sensors-24-07830-f005:**
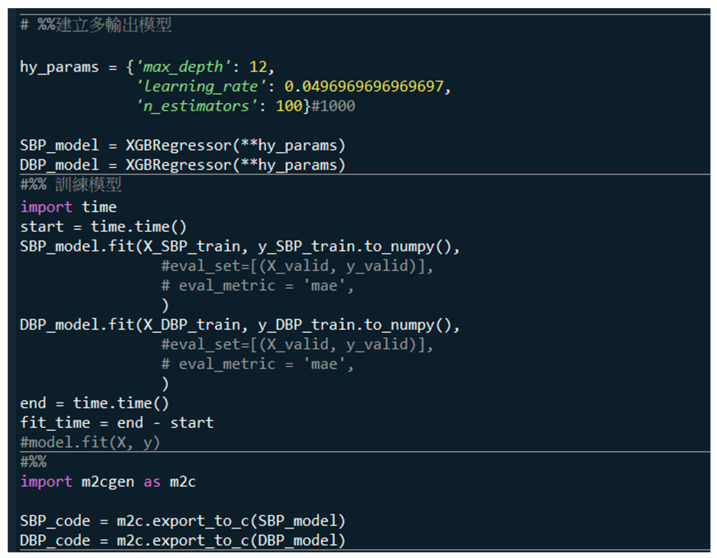
The last three comments are used to convert the estimation models of SBP and DBP as the C-code models.

**Figure 6 sensors-24-07830-f006:**
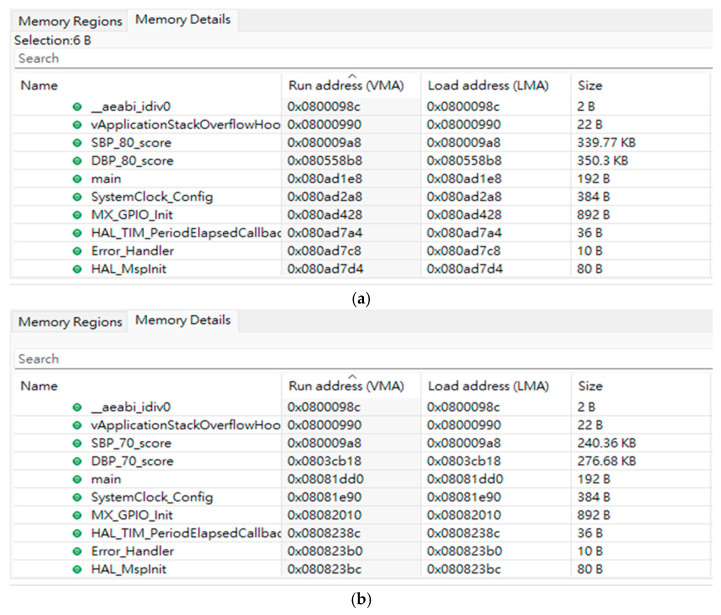
The flash memory information of STM32F756ZG microcontroller. (**a**) XGBoost with 80 trees; the memories of SBP and DBP models are 339.77 Kbytes and 350.3 Kbytes, respectively. (**b**) XGBoost with 70 trees; the memories of SBP and DBP models are 240.36 Kbytes and 276.68 Kbytes, respectively.

**Table 1 sensors-24-07830-t001:** The metrics of five-fold crossing-validation for PCCs of SBP and DBP estimations with different numbers of trees, 100, 90, 80, 70, 60, and 50.

Numbers of Trees	SBP	DBP
100	0.78727 ± 0.013	0.77318 ± 0.021
90	0.78843 ± 0.013	0.77609 ± 0.019
80	0.79089 ± 0.012	0.77926 ± 0.018
70	0.79293 ± 0.016	0.78116 ± 0.016
60	0.79307 ± 0.015	0.78251 ± 0.015
50	0.79118 ± 0.013	0.78046 ± 0.014

**Table 2 sensors-24-07830-t002:** The metrics of five-fold crossing-validation for MAEs of SBP and DBP estimations with different numbers of trees, 90, 80, 70, 60, and 50.

Numbers of Trees	SBP (mmHg)	DBP (mmHg)
90	7.52 ± 0.18	4.88 ± 0.11
80	7.63 ± 0.20	5.01 ± 0.11
70	7.99 ± 0.25	5.35 ± 0.09
60	8.96 ± 0.46	6.16 ± 0.13
50	11.31 ± 0.22	7.82 ± 0.16

**Table 3 sensors-24-07830-t003:** The metrics of BP estimation under server-based computing and edge computing.

	Server-Based Computing		Edge Computing	
	SBP	DBP	SBP	DBP
PCC	0.80	0.81	0.73	0.78
ME (mmHg)	2.64 ± 9.71	1.52 ± 6.32	2.2 ± 10.9	1.87 ± 6.79
MAE (mmHg)	7.63 ± 0.20	5.01 ± 0.11	8.58 ± 7.2	5.27 ± 4.66

**Table 4 sensors-24-07830-t004:** Comparative result of various methods using the ECG or BCG as the proximal reference, and IPG or PPG as the distal reference for the cuffless blood pressure measurement.

Reference	PTT Signals (Sensor Placement)	PCC	
[39]	BCG (foot) and BPW (finger)	SBP: 0.70 DBP: 0.66	NA
[40]	BCG (chair)	SBP: 0.755 DBP: 0.532	ME SBP: 0.93 ± 6.24 mmHg DBP: 0.21 ± 5.42 mmHg
[41]	BCG (foot) and PPG (foot)	NA	RMSE SBP: 11.8 ± −1.6 mmHg DBP: 7.6 ± −0.5 mmHg,
[42]	ECG and IPG (arm)	SBP: 0.700 DBP: 0.450	NA
[15]	BCG (foot) and PPG (foot)	SBP: 0.775 DBP: 0.532	RMSE SBP: 6.7 ± 1.6 mmHg DBP: 4.8 ± 1.47 mmHg
[16]	BCG (foot) and IPG (foot)	SBP: 0.754 DBP: 0.533	RMSE SBP: 7.3 ± 2.1 mmHg DBP: 4.5 ± 1.8 mmHg
Proposed method: Server-based Computing	BCG (foot) and IPG (foot)	SBP: 0.80 DBP: 0.81	ME SBP: 2.64 ± 9.71 mmHg DBP: 1.52 ± 6.32 mmHg
Proposed method: Edge Computing	BCG (foot) and IPG (foot)	SBP: 0.73 DBP: 0.78	ME SBP: 2.2 ± 10.9 mmHg DBP: 1.87 ± 6.79 mmHg

## Data Availability

Data are contained within the article.

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
