# Peer review of "Using a Bodily Weight-Fat Scale for Cuffless Blood Pressure Measurement Based on the Edge Computing System"

_sensors, 2024, doi:10.3390/s24237830_

Round 1

Reviewer 1 Report

Comments and Suggestions for Authors

The aim of this study is to realize an edge computing for the BP measurement. This paper is easy to follow. Some issues should be addressed as follows.

1.     In the abstract, more details of main contributions and novelty should be emphasized.

2.     In the introduction, the background about the world’s population should be simplified. Besides, more related works should be introduced and compared with their strengths.

3.     The format of abbreviations should be checked and revised, i.e., systolic blood pressure (SBP) has appeared several times in the introduction.

4.     As the background of this paper includes edge computing, recent high quality works on edge computing should be introduced, such as DRL-based offloading for computation delay minimization in wireless-powered multi-access edge computing, IEEE TCOM.

5.     Some references are outdated and should be updated.

6.     The motivations and main contributions should be better summarized in the introduction.

7.     Why the BCG circuit is designed as figure 2? Similarly, why the IPG circuit is designed as figure 3? Are these two circuits designed by the authors and could be viewed as the main contributions?

8.     The referring equations should be revised, “Eq. 1 is a Z transfer function” should be revised as “(1) represents the Z transfer function”.

9.     Codes are not suggested to be included as figure 5 in the paper. Authors are suggested to provide the pseudocode.

10.  Important results or findings should be summarized in conclusion.

Author Response

To the Reviewer:

We sincerely thank the Reviewer for providing valuable comments that have helped improve this manuscript. In this revised version, the corrections/modifications have been marked in red. It is our sincere hope that these revisions will enhance the readability and strengthen the content of the manuscript to meet the high standards of this prestigious journal.

Comments and Suggestions for Authors

The aim of this study is to realize an edge computing for the BP measurement. This paper is easy to follow. Some issues should be addressed as follows.

  1. In the abstract, more details of main contributions and novelty should be emphasized.

ANS: We modify the mention in Abstract chapter.

Abstract: Blood pressure (BP) measurement is a major physiological information for people with the cardiovascular diseases, like as hypertension, heart failure, and atherosclerosis. Moreover, elders and patients with kidney disease, and diabetes mellitus also are suggested to measure BP at every day. The cuffless BP measurement has been developed in the past 10 years, which benefit is comfortable to users. Now, ballistocardiogram (BCG) and impedance plethysmogram (IPG) could be used to perform the cuffless BP measurement. Thus, the aim of this study is to realize an edge computing for the BP measurement in real time, which includes measurement of BCG and IPG signals, digital signal process, feature extraction, and BP estimation by machine learning algorithm.  This system measured BCG and IPG signals from a bodily weight-fat scale with the self-made circuits. The signals were filtered to reduce the noise and segmented by 2 seconds. Then, we proposed a flowchart to extract the parameter, pulse transit time (PTT), within each segment. The feature including two calibration-based parameters and one calibration-free parameter was used to estimate BP with XGBoost. In order to realize the system in STM32F756ZG NUCLEO development board, we limited the hyper parameters of XGBoost model, including maximum depth (max_depth) and tree number (n_estimators). Results show that the error of systolic blood pressure (SBP) and diastolic blood pressure (DBP) in the server-based computing are 2.64 ± 9.71mmHg and 1.52 ± 6.32 mmHg, and in the edge computing are 2.2 ± 10.9 mmHg and 1.87 ± 6.79 mmHg. This proposed method significantly enhances the feasibility of bodily weight-fat scale in the BP measurement for effective utilization in mobile health applications.

  1. In the introduction, the background about the world’s population should be simplified. Besides, more related works should be introduced and compared with their strengths.

ANS: We delete two sentences about population in Taiwan and Japan.

Recently, the lifespans of the world’s population are increasing, and society is gradually aging. According to the report of the United Nations [1], the number of elderly people (over 65) in the world in 2019 was 703 million, and this is estimated to double to 1.5 billion by 2050. From 1990 to 2019, the proportion of the global population over 65 years old increased from 6% to 9%, and the proportion of the elderly population is expected to further increase to 16% by 2050. Thus, the cost of home care for elders will significantly increase. In homecare, the monitoring of elderly people living alone is a major issue. Some studies have proposed that engaged elders could use sophisticated digital technologies for self-monitoring and self-care [2]. Large amounts of data and information are measured from such digital apparatuses, relating to the past, present, and future physical and mental health or condition of an individual, which can be analyzed to extract knowledge for improving the public health and providing basic services [3].

  1. The format of abbreviations should be checked and revised, i.e., systolic blood pressure (SBP) has appeared several times in the introduction.

ANS: We have carefully checked these abbreviations and modified in this manuscript.

  1. As the background of this paper includes edge computing, recent high quality works on edge computing should be introduced, such as DRL-based offloading for computation delay minimization in wireless-powered multi-access edge computing, IEEE TCOM.

ANS: We add this reference in Introduction chapter.

An edge computing system combines with 5G communication and modern computing techniques, which can perform the real-time monitor or measurement in a mobile device. Zheng et al. studied the wireless-powered multi-access edge computing network, where wireless devices conducted either local computing or task offloading for their undividable computation tasks [17-1]. Now, some studies review its approaches, opportunities, and challenges in the smart health [18,19]. The modern computing techniques focus on deep learning and ensemble machine learning algorithms, which can be executed in a microcontroller. Chin et al. used an Arduino Nano 33 BLE Sense development board to classify the emergency vehicle sirens with an EfficientNet-based ensemble model [20]. Rahman et al. developed a deep learning model to detect COVID-19 symptoms based on a smartphone [21]. Goossens et al. proposed a state-of-the-art algorithms of edge computing for the real-time BP estimation and ECG compression [22]. However, this study focused on the power consumption and executing time. Many studies proposed the methods of cuffless BP measurement but few researches studied the cuffless BP measurement in an edge computing environment. Bernard et al. used the cloud serve to collect PPG signal measured by different bedside monitors and build the deep learning and machine learning models to estimate BP [22-1]. Sun et al. also used PPG signal from MIMIC-IV database to estimate BP with five convolutional neural networks in Arduino Nano 33 BLE development board. Only AlexNet has the least loss in performance about 8%. [22-2]. Ahmed et al. also used MIMIC-IV database to build edge computing system with six machine learning models in ESP32 Wrover Board [22-3]. The mean absolute error (MAE) of SBP and DBP were14.08±17.82 mmHg and 6.85±9.16 mmHg, respectively. However, the edging computing technique should include the sensors, signal measurement, signal process, feature extraction, building model, adjustment of hyperparameter, imbedding in a microcontroller, and evaluation of performance. The previous studies did not fully fit the requirement of edge computing system.

  1. Some references are outdated and should be updated.

ANS: We have added some new studies as the references in Introduction chapter.

  1. Zheng, K.; Jiang, G.; Liu, X.; Chi, K.; Yao, X.; Liu, J. DRL-based offloading for computation delay minimization in wireless-powered multi-access edge computing. IEEE Transactions on Communications 2023, 71, pp. 1755-1770.
  2. Bernard, D.; Msigwa, C.; Yun, J. Toward IoT-based medical edge devices: PPG-based blood pressure estimation application. IEEE Internet of Things Journal 2023, 10, 5240-5255. doi: 10.1109/JIOT.2022.3222477.
  3. Ahmed, K.; Hassan, M. tinyCare: a tinyML-based low-cost continuous blood pressure estimation on the extreme edge. IEEE 10th International Conference on Healthcare Informatics, Rochester, MN, USA, 2022, 264-275, doi: 10.1109/ICHI54592.2022.00047.
  4. Sun, B.; Bayes, S.; Abotaleb, A. M.; Hassan, M. The case for tinyML in healthcare: CNNs for real-time on-edge blood pressure estimation, Proceedings of the 38th ACM/SIGAPP Symposium on Applied Computing, 2023, 629 – 638. https://doi.org/10.1145/3555776.3577747

  1. The motivations and main contributions should be better summarized in the introduction.

ANS: We add a paragraph to mention the major contributions in Introduction chapter.

The main contributions of this study are summarized as follows:

  • This study uses the self-made circuits to measure BCG and IPG signals from bodily weight-fat scale. These signals were filtered and segmented to extract PTT parameter.
  • This study proposes an operation procedure to extract PTT parameter in edge computing system.
  • Our proposed models for SBP and DBP estimations in this study are tested on the STM32F756ZG NUCLEO development board. This system could perform the cuffless BP measurement in real time.
  • This study verifies the performances of server-based computing and edge computing. The edge computing has the loss in performance about 8%.

  1. Why the BCG circuit is designed as figure 2? Similarly, why the IPG circuit is designed as figure 3? Are these two circuits designed by the authors and could be viewed as the main contributions?

ANS: Many thanks for reviewer comment. BCG and IPG circuit are designed and made by our team. This contribution is listed in the main contribution in Introduction chapter. Moreover, we modify the mention for Figure 2 and 3.

Figure 2 shows the BCG schematic which used two strain gauges to detect BCG signal.

Figure 3 shows the IPG schematic which used four electrode pads to fed the fixed alternated current into the body, and extract the potential voltage of body.  

  1. The referring equations should be revised, “Eq. 1 is a Z transfer function” should be revised as “(1) represents the Z transfer function”.

ANS: We modify these sentences.

The infinite impulse response filters were used to remove the noises of BCG and IPG. We used the recursive approach to filter signals in real time. For BCG signal, Eq. (1) represents the Z transfer function of fourth-order Butterworth high pass filter (0.5 Hz of cutoff frequency), and Eq. (2) represents the Z transfer function of fourth-order Butterworth low pass filter (20 Hz of cutoff frequency). For IPG signal, Eq. (3) represents the Z transfer function of second-order Chebyshev high pass filter (0.3 Hz of cutoff frequency), and Eq. (4) represents the Z transfer function of fourth-order Chebyshev low pass filter (10 Hz of cutoff frequency).

  1. Codes are not suggested to be included as figure 5 in the paper. Authors are suggested to provide the pseudocode.

ANS: Many thanks for reviewer comment. Figure 5 shows the how to use m2cgen tool to convert XGBoost by Python code to C code. According to reviewer 2 comment, we add some sentences to describe these Python codes.

2.3.2. Python to C codes

In server-based computing system, the format of trained model could be .pb, .onnx, .ckpt, .tflite or .h5. However, these formats cannot be directly imported into the microcontroller, so the special conversion of the format is required. When training the model, we use the m2cgen (Model 2 Code Generator) function library to convert the XGBoost model as C code, such as shown in Fig. 5. The last three comments convert the estimation models of SBP and DBP as C-code models. First, we import the m2cgen tool, “import m2cgen as m2c”. Then, we export C-code models of SBP and DBP estimation as SBP_code and DBP_code, “SBP_code = m2c.export_to_c(SBP_model)” and “DBP_code=m2.export_to_c(DBP_model).

  1. Important results or findings should be summarized in conclusion.

ANS: We modify the texts in Conclusions chapter.

  1. Conclusions

In this study, we proposed an operation procedure to use the bodily weight-fat scale for the real-time cuffless BP measurement based on STM32F756ZG NUCLEO development board. The self-made circuits were used to measure BCG and IPG signals from the bodily weight-fat scale. These signals were filtered and segmented. We proposed a method to extract PTT from BCG and IPG signals in real time. Then, we limited the hyper parameters of XGBoost to reduce memory size of models for SBP and DBP estimations which could be executed on STM32F756ZG NUCLEO development board. Moreover, we also compared the performances of server-Based computing and edge computing. The edge computing has 8% loss in performances. If the calibration-based or calibration-free parameters of PTT can be improved, the accuracy of real-time cuffless BP measurement will arrive the A grade for IEEE Standard for Wearable, Cuffless Blood Pressure Measuring Devices. This advancement would support the convenient BP monitoring in daily life, and facilitate the progress of mobile health in the future.

Reviewer 2 Report

Comments and Suggestions for Authors

This paper introduces an  edge-computing-based cuffless blood pressure measurement system that leverages pulse transit time (PTT) estimates from ballistocardiogram (BCG) and impedance plethysmogram (IPG) signals, using a weight-fat scale as the primary hardware. This approach not only enhances the functionality of weight-fat scales in measuring blood pressure but also holds significant promise for mobile health (mHealth) applications. The system processes BCG and IPG signals through the XGBoost algorithm and implements edge computation on the STM32F756ZG NUCLEO development board. This setup reduces reliance on external power sources, improving portability and practicality. Experimental results demonstrate that the system's estimation errors for both systolic blood pressure (SBP) and diastolic blood pressure (DBP) remain within acceptable ranges, indicating its effectiveness for mHealth integration and laying the groundwork for future self-sustaining sensor networks.

  1. The abstract would benefit from a more detailed description of the innovative method, focusing on the edge computing approach and signal processing techniques used. A more concise description of traditional methods would allow readers to more easily understand the novelty and advantages of the proposed system.

  2. Some figures, especially Figure 2, could be improved for better readability. Enhancing image quality, resolution, and labeling would aid readers in understanding the technical aspects presented.

  3. The introduction section lacks sufficient references to related works. Additional citations could strengthen the context and background for the study, allowing readers to see how this work compares with previous research.

  4. The explanation of code, particularly in Figure 5 on page 7, is insufficiently detailed. Expanding the accompanying text to clarify key aspects of the code would improve comprehension for readers seeking to replicate or build upon this work.

  5. A new table summarizing key data could be beneficial to present experimental results clearly. Furthermore, ensure that units are included for all tables where applicable (e.g., Table 2).

  6. A conclusion section summarizing the main contributions, findings, and implications of the study is currently missing. Adding this section would provide a systematic summary and reinforce the paper’s significance and practical applications.

Author Response

To the Reviewer:

We sincerely thank the Reviewer for providing valuable comments that have helped improve this manuscript. In this revised version, the corrections/modifications have been marked in red. It is our sincere hope that these revisions will enhance the readability and strengthen the content of the manuscript to meet the high standards of this prestigious journal.

Comments and Suggestions for Authors

This paper introduces an  edge-computing-based cuffless blood pressure measurement system that leverages pulse transit time (PTT) estimates from ballistocardiogram (BCG) and impedance plethysmogram (IPG) signals, using a weight-fat scale as the primary hardware. This approach not only enhances the functionality of weight-fat scales in measuring blood pressure but also holds significant promise for mobile health (mHealth) applications. The system processes BCG and IPG signals through the XGBoost algorithm and implements edge computation on the STM32F756ZG NUCLEO development board. This setup reduces reliance on external power sources, improving portability and practicality. Experimental results demonstrate that the system's estimation errors for both systolic blood pressure (SBP) and diastolic blood pressure (DBP) remain within acceptable ranges, indicating its effectiveness for mHealth integration and laying the groundwork for future self-sustaining sensor networks.

  1. The abstract would benefit from a more detailed description of the innovative method, focusing on the edge computing approach and signal processing techniques used. A more concise description of traditional methods would allow readers to more easily understand the novelty and advantages of the proposed system.

ANS: We modify the texts in Abstract chapter.

Abstract: Blood pressure (BP) measurement is a major physiological information for people with the cardiovascular diseases, like as hypertension, heart failure, and atherosclerosis. Moreover, elders and patients with kidney disease, and diabetes mellitus also are suggested to measure BP at every day. The cuffless BP measurement has been developed in the past 10 years, which benefit is comfortable to users. Now, ballistocardiogram (BCG) and impedance plethysmogram (IPG) could be used to perform the cuffless BP measurement. Thus, the aim of this study is to realize an edge computing for the BP measurement in real time, which includes measurement of BCG and IPG signals, digital signal process, feature extraction, and BP estimation by machine learning algorithm.  This system measured BCG and IPG signals from a bodily weight-fat scale with the self-made circuits. The signals were filtered to reduce the noise and segmented by 2 seconds. Then, we proposed a flowchart to extract the parameter, pulse transit time (PTT), within each segment. The feature including two calibration-based parameters and one calibration-free parameter was used to estimate BP with XGBoost. In order to realize the system in STM32F756ZG NUCLEO development board, we limited the hyper parameters of XGBoost model, including maximum depth (max_depth) and tree number (n_estimators). Results show that the error of systolic blood pressure (SBP) and diastolic blood pressure (DBP) in the server-based computing are 2.64 ± 9.71mmHg and 1.52 ± 6.32 mmHg, and in the edge computing are 2.2 ± 10.9 mmHg and 1.87 ± 6.79 mmHg. This proposed method significantly enhances the feasibility of bodily weight-fat scale in the BP measurement for effective utilization in mobile health applications.

  1. Some figures, especially Figure 2, could be improved for better readability. Enhancing image quality, resolution, and labeling would aid readers in understanding the technical aspects presented.

ANS: We modify the text size and add the resolution of figure.

  1. The introduction section lacks sufficient references to related works. Additional citations could strengthen the context and background for the study, allowing readers to see how this work compares with previous research.

ANS: We add this reference in Introduction chapter.

An edge computing system combines with 5G communication and modern computing techniques, which can perform the real-time monitor or measurement in a mobile device. Zheng et al. studied the wireless-powered multi-access edge computing network, where wireless devices conducted either local computing or task offloading for their undividable computation tasks [17-1]. Now, some studies review its approaches, opportunities, and challenges in the smart health [18,19]. The modern computing techniques focus on deep learning and ensemble machine learning algorithms, which can be executed in a microcontroller. Chin et al. used an Arduino Nano 33 BLE Sense development board to classify the emergency vehicle sirens with an EfficientNet-based ensemble model [20]. Rahman et al. developed a deep learning model to detect COVID-19 symptoms based on a smartphone [21]. Goossens et al. proposed a state-of-the-art algorithms of edge computing for the real-time BP estimation and ECG compression [22]. However, this study focused on the power consumption and executing time. Many studies proposed the methods of cuffless BP measurement but few researches studied the cuffless BP measurement in an edge computing environment. Bernard et al. used the cloud serve to collect PPG signal measured by different bedside monitors and build the deep learning and machine learning models to estimate BP [22-1]. Sun et al. also used PPG signal from MIMIC-IV database to estimate BP with five convolutional neural networks in Arduino Nano 33 BLE development board. Only AlexNet has the least loss in performance about 8%. [22-2]. Ahmed et al. also used MIMIC-IV database to build edge computing system with six machine learning models in ESP32 Wrover Board [22-3]. The mean absolute error (MAE) of SBP and DBP were14.08±17.82 mmHg and 6.85±9.16 mmHg, respectively. However, the edging computing technique should include the sensors, signal measurement, signal process, feature extraction, building model, adjustment of hyperparameter, imbedding in a microcontroller, and evaluation of performance. The previous studies did not fully fit the requirement of edge computing system.

  1. The explanation of code, particularly in Figure 5 on page 7, is insufficiently detailed. Expanding the accompanying text to clarify key aspects of the code would improve comprehension for readers seeking to replicate or build upon this work.

ANS: We add some sentences to describe how to use m2cgen tool in 2.3.2. Chapter.

2.3.2. Python to C codes

In server-based computing system, the format of trained model could be .pb, .onnx, .ckpt, .tflite or .h5. However, these formats cannot be directly imported into the microcontroller, so the special conversion of the format is required. When training the model, we use the m2cgen (Model 2 Code Generator) function library to convert the XGBoost model as C code, such as shown in Fig. 5. The last three comments convert the estimation models of SBP and DBP as C-code models. First, we import the m2cgen tool, “import m2cgen as m2c”. Then, we export C-code models of SBP and DBP estimation as SBP_code and DBP_code, “SBP_code = m2c.export_to_c(SBP_model)” and “DBP_code=m2.export_to_c(DBP_model).     

  1. A new table summarizing key data could be beneficial to present experimental results clearly. Furthermore, ensure that units are included for all tables where applicable (e.g., Table 2).

ANS: Table 2 shows the PCC (Pearson correlation coefficient) metrics of SBP and DBP estimations. Thus, there are not the unit. The results of this study show in Table 4 which includes the performances of server-based computing and edge computing.

Table 4. The metrics of BP estimation under server-based computing and edge computing.

Server-based Computing

Edge Computing

SBP

DBP

SBP

DBP

PCC

0.80

0.81

0.73

0.78

ME (mmHg)

2.64 ± 9.71

1.52 ± 6.32

2.2 ± 10.9

1.87 ± 6.79

MAE (mmHg)

7.63 ± 0.20

5.01 ± 0.11

8.58 ± 7.2

5.27 ± 4.66

  1. A conclusion section summarizing the main contributions, findings, and implications of the study is currently missing. Adding this section would provide a systematic summary and reinforce the paper’s significance and practical applications.

ANS: We modify the mention in Conclusions chapter.

  1. Conclusions

In this study, we proposed an operation procedure to use the bodily weight-fat scale for the real-time cuffless BP measurement based on STM32F756ZG NUCLEO development board. The self-made circuits were used to measure BCG and IPG signals from the bodily weight-fat scale. These signals were filtered and segmented. We proposed a method to extract PTT from BCG and IPG signals in real time. Then, we limited the hyper parameters of XGBoost to reduce memory size of models for SBP and DBP estimations which could be executed on STM32F756ZG NUCLEO development board. Moreover, we also compared the performances of server-Based computing and edge computing. The edge computing has 8% loss in performances. If the calibration-based or calibration-free parameters of PTT can be improved, the accuracy of real-time cuffless BP measurement will arrive the A grade for IEEE Standard for Wearable, Cuffless Blood Pressure Measuring Devices. This advancement would support the convenient BP monitoring in daily life, and facilitate the progress of mobile health in the future.

Round 2

Reviewer 1 Report

Comments and Suggestions for Authors

Authors have addressed my previous concerns.

Author Response

Many thanks for reviewer's comment.